# Recycling of Plastic Food Packages: A Case Study with Finnish University Students

Jarmo Alarinta *, Margit Närvä and Gun Wirtanen

Seinäjoki University of Applied Sciences, P.O. Box 412, FI-60101 Seinäjoki, Finland
* Correspondence: jarmo.alarinta@seamk.fi; Tel.: +358-40-830-2450

**Abstract:** Recycling, depositing, and proper discarding of plastics are significant means to reduce plastics in the environment. The purpose of this study was to monitor both the type and amount of plastic food packages recycled, reused, and discarded in Finnish households with at least one university student. The participating students came from various universities of applied sciences. They participated in courses related to sustainable food systems at Seinäjoki University of Applied Sciences. In total, 785 approved participants from 363 households took part in one-week monitoring. The focus was to quantify the number of food packages used and specify how the respondents handled the food packages after use. This study shows that the recycling rate of plastic packages in Finnish households was 61%. Bigger households produce less packaging waste per person than one- or two-person households. Furthermore, the recycling activity in single-person households was lower than for two- and three-person households. The Finnish deposit system for drink packages encourages people to recycle packages. This reduces municipal waste. Recycling requires knowledge of the plastic material used in food packages.

**Keywords:** food packaging materials; recycling of plastic food packages; food package waste; package deposit; food safety





## 1. Introduction

The directive (EU) 2018/852 [1] amending the packaging and packaging waste directive 94/62/EC [2] states that the European Union (EU) Member States should focus on minimizing the environmental impact of packages and packaging waste. More packaging waste has been recycled since the EU-wide targets for packages of paper, cardboard, glass, plastic, ferrous metals, aluminium, and wood were introduced in the 1990s [1,2]. Plastic packages consist mainly of bottles, cans, wraps, films, bags, and pouches [1]. The municipal waste generated in the EU accounts for 7–10% of the total waste. The municipal waste treatment should be based on efficient collection and functional sorting systems. Furthermore, the reuse and recycling, including package deposits of packaging material as well as discarding packaging waste, must be enhanced [1]. Several countries encourage consumers to separate packaging waste from materials to be recycled. The revision on waste handling includes strict recycling targets for packaging materials [1]. In order to enable a functioning circular economy, the directive contains updated measures designed to prevent the production of packaging waste [1].

The packages protect food from various post-process hazards after processing during storage and distribution. In the recycling of packages, measures should be taken to ensure the hygiene of these packages. Single-use packages are used to maximize the safety of food products. The multilayered laminates are used in food packages to exclude oxygen, generally enabling a longer shelf-life of the packaged food [3]. The laminates in food packages have received significant interest due to their versatility, functionality, and convenience, as they combine the benefits of each layer with improved barrier properties, mechanical integrity, and functional features [4]. Protective polymers as gas-resistant barriers, e.g.,

ethylene vinyl alcohol (EVOH) and polyamide (PA), are laminated on moisture-resistant or sealable polymers [5]. Billions of kilograms of multilayer membranes, which are relatively pure of almost standardized composition, are produced annually. Unfortunately, there are no commercially available techniques to deconstruct these multilayer films into pure, recyclable polymers [5].

A review of the environmental impact of food packaging has led to several global directives on sustainability. Here, the direct effects of materials have been in focus [6]. According to Heller et al. [7], the growing awareness of the effects of food waste requires the calibration of the environmental effects of packages to include these indirect effects in food waste. Food packaging must be considered a part of the environmental impact of food production and consumption [1]. Heller et al. [7] used thirteen food products in consistent life cycles with typical package analysis to demonstrate the impact of food waste on total greenhouse gas emissions and cumulative energy demands. This demonstration showed the importance of food with proper packaging options to reduce the overall environmental effect [7]. Costello et al. [8] showed in their scenarios that recycling packaging materials after the removal of edible food waste from the packages is the most effective way to reduce both greenhouse gas emissions and energy used. Wikström et al. [9] presented connections between packaging systems and food waste. These factors were (1) identifying and acquiring specific information on packaging affecting food waste; (2) understanding the total environmental load on the products/packages considering both product protection and preservation as well as environmental footprint mitigation; (3) developing methods addressing environmental footprint assessments; (4) improving packaging design to reduce food waste; and (5) analysing stakeholder incentives to reduce both food waste and food loss. An optimized, feasible packaging solution affects the reduction of both food and package waste [9,10].

The term "reuse" covers packages used in the original form. Recycling covers the reworking of material into packaging material. All packaging materials are technically recyclable, but the economics of recycling favours containers made of an easily identifiable pure material, e.g., glass, metal, high-density polyethylene (PE-HD), PET, and paper/carton. The suitability of some plastics in recycling depends on the fact that they are inert and absorb fewer "impurities" from the food. Both PET and polyvinyl chloride (PVC) are more inert than polystyrene (PS) and PE-HD; thus, PVC and PET are more suitable for recycling [11,12]. The recycling plants have evolved significantly. Modern automatically operated sorting plants can distinguish single polymer plastic fractions of packages returned [13]. Plastic waste can also be converted into energy and other valuable products by pyrolysis. In the study by Miandad et al. [14], the catalytic pyrolysis of PS produced higher liquid oil (70 and 60%) than PP (40%) and PE (40 and 42%). Active sorting by the consumers is required to enable a proper flow of complex waste. The recycling of plastic material is particularly complicated because several types of polymers are glued together [13].

The materials with international package codes are consumer friendly, especially in the recycling of plastics [15]. Furthermore, e.g., bio-based polyester polymers biodegrade slowly [16]. Non-synthetic materials, e.g., polysaccharides and proteins, are degrading rapidly and can thus be considered useful processing aids and fillers in biopolymers [17]. Note that polyethylene terephthalate (PETE or PET) plastics are one decade less permeable than other plastics. The EVOH and PA barriers are denser than PET, and thus, they provide better protective gas packages with lower material consumption. The recycling targets for the EU in 2030 are so demanding that structural changes are currently needed for plastic food packages [6].

Only a limited number of studies have been carried out based on consumer solutions in the recycling system. The survey on Finnish recycling behavior by Reijonen et al. [18] supports the view that social norms had an insignificant relationship. The recycling of plastic packaging has been boosted by national public plastic recycling campaigns. This has increased the number of waste points and requirements for packaging manufacturers. The

deposit system might also encourage people to recycle packages and thus reduce packages ending up in municipal waste.

Cavaliere et al. [19] investigated the consumers' willingness to avoid the use of plastics in food packages based on a structural equation model. The findings revealed that environmental and health concerns are key drivers in the avoidance of plastics. It is, however, difficult to replace plastic completely because plastics have excellent barrier properties. Furthermore, these materials are light and durable [20]. The recovery of material could be improved with easy, proper sorting and separation instructions on the packages [19] because consumers may choose not to sort packages due to confusing instructions. In Nigeria, Aikowe and Mazancová [21] studied the recycling intentions of plastic materials for university students. Experienced behavioural control had a great impact on respondents sorting plastic waste. Subsequently, standards, e.g., environmental awareness, voluntary activities, and research programmes, were found relevant for sorting plastic waste. Cavaliere et al. [19] examined the consumers' decision-making to find out what affects refraining purchases of food products containing several plastic packages and plastic water bottles, etc.

Consumer choices will determine the success of the current recycling system. The purpose of this paper was to investigate various plastics and amounts of food packages wasted in Finnish households with at least one university student. The aim was to determine whether the total amount of waste produced and the sorting activity in the household were affected. The sustainability in the recycling of packages must be improved on global, regional, and local levels. The industrial transition to the more sustainable use of plastic material has already begun, but the process will take years to become feasible both in industrial operations and in the economy [20]. The main purpose of the study was to quantify both the recycling and sorting of food package waste in households with university students who attended courses related to sustainable food systems in 2019–2020.

## 2. Results

### 2.1. Background Information of Households

The total accepted number of Excel sheets with monitoring results was 363 households. Four replies were rejected. The reason for rejection was that these reports did not indicate specific quantities of packages. The rejected answers were equivalent to approx. 1% of all returned forms. The full-time students in Finnish households were studying either in the daytime or alongside their work. The students lived in apartments with a kitchen or at least a kitchenette, and they were either living alone or together with other household members, e.g., parents, a spouse, children, or one or more friends. The total number of persons in the participating households was 785. These student households had at least one representative who participated in sustainable food systems courses in 2019–2020 from various universities of applied sciences. Seinäjoki University of Applied Sciences arranged these courses. In Table 1, the respondents, i.e., students and other household members, were divided by the size of households (Table 1a), by the age of respondents (Table 1b), and by the number of respondents in various-sized households (Table 1c). A big part of the respondents were young adults. This is understandable because this study focused on university students. The biggest part was single-person households. This size of household is growing with urbanization [22]. The household sizes in this study represent reasonably well the average distribution of households in Finland [22].

**Table 1.** The respondents divided by the (**a**) size of households, (**b**) age of the persons living in the various-sized households, and (**c**) number of respondents in all households.

| Number of Households | % | Number of Persons | % | Number of Persons | % |
|---|---|---|---|---|---|
| Size of Household | | Age | | Respondents by Household Size | |
| 1 | 137 | 38 | <12 | 126 | 16 | 1 | 137 | 17 |
| 2 | 124 | 34 | 12–18 | 51 | 6 | 2 | 248 | 32 |
| 3 | 42 | 12 | 19−24 | 229 | 29 | 3 | 126 | 16 |
| 4 | 39 | 11 | 25−30 | 129 | 16 | 4 | 156 | 20 |
| ≥5 | 21 | 6 | 31−40 | 122 | 16 | ≥5 | 118 | 15 |
| | | | 41−50 | 77 | 10 | | | |
| | | | >50 | 51 | 6 | | | |
| Total | 363 | 100 | Total | 785 | 100 | Total | 785 | 100 |
| | (a) | | | (b) | | | (c) | |

## 2.2. Amount and Type of the End-Use of the Food Packages

The total number of plastic entries was 9009 (Table 2) during the monitoring period, i.e., an average of 11 entries/person/week or 1.6 entries/person/day. In Table 2, the entries were divided based on end-use and type of polymers. Five polymers commonly used in packaging are (1) polyethylene terephthalate, PET, (2) high-density polyethylene, PE-HD, (3) low-density polyethylene, PE-LD, (4) polypropylene, PP, and (5) polystyrene, PS. Another plastic grade, which is typical, is a multilayer laminate with barriers. The numbers refer to the plastic identification codes [15]. The recycling label facilitates the identification of different plastic grades, but this requires attention. It can be stated that the distribution of plastic packages was fairly even. PET entry was the most common due to its use in soft drink and spring water bottles as well as in convenience food trays. In this study, the total recycling rate of plastic packages corresponds to approximately half of all plastic packages (58%). The rest of the packages were discarded as municipal waste, i.e., as mixed waste or as energy waste (Table 2). Most of the mixed waste (96%) collected in Finland is exploited as energy with low effect on climate [23]. In addition to this, Jeswani et al. [24] stated that chemical recycling of mixed plastic waste (MPW) via pyrolysis would affect climate change less, up to 50% less. This is based on the burning option in the energy recovery life cycle.

**Table 2.** Quantitative summary of used plastic packages based on both end-use and type of polymers.

| | Plastic [a] | | | | | | | | | | | | | |
|---|---|---|---|---|---|---|---|---|---|---|---|---|---|---|
| Target/Amount | PET Bottle/Tray | % | PE-HD Bottle/Tray | % | PE-LD Film | % | PP Pot | % | PS Pot | % | O Tray | % | Total pcs | Share % |
| Deposit | 1584 | 83 | 1 | 0 | 5 | 0 | - | - | 8 | 0 | 1 | 0 | 1683 | 18 |
| Recycling | 223 | 12 | 403 | 44 | 748 | 47 | 858 | 52 | 729 | 45 | 635 | 50 | 3794 | 40 |
| Municipal waste | 104 | 5 | 516 | 56 | 851 | 53 | 806 | 48 | 894 | 55 | 643 | 50 | 4032 | 42 |
| Total | 1911 | 100 | 920 | 100 | 1604 | 100 | 1664 | 100 | 1631 | 100 | 1279 | 100 | 9509 | 100 |

[a] PET = polyethylene terephthalate, PE-HD = high-density polyethylene, PE-LD = low-density polyethylene, PP = polypropylene, PS = polystyrene & O = multi-layer laminates.

Depending on the material, the recycling rates varied significantly (Table 2). Pure PET is perfectly recyclable and reusable as food packages after flake washing. There is a national deposit refund system for drink and beverage packages of plastic, glass, and metal in Finland. The refund for plastic beverage bottles equals 10, 20, or 40 cents/package, which encourages the recycling of packages. This result is also visible in this study, in which only five percent of the PET packages ended up in energy production (Table 2). The recycling rate for PE packages is only 44–47%, depending on PE quality. PE-HD, PE-LD, and PP are purely recoverable as raw materials for various structural products, excluding food packages. The recycling rate of PP is 52%. Pure PS and multilayer laminates are utilized in recycled plastic profiles, which are used in environmental constructions such as noise fences

and grazing posts [25]. MPW packages, which consist of very heterogeneous fractions and heavily contaminated materials, are a challenge in the recycling system. Furthermore, the plastic packages end up in MPW, which must be mechanically sorted [24]. The study by Antonoplous et al. [26] showed that the easiest polymer to sort was HD-PE (93% average purity), followed by PET (91%), whereas PS (71%) was the hardest to sort mechanically.

### 2.3. Sorting of End-Use Packages in Households

In this study, the sizes of households with university students represent reasonably well the average distribution of households in Finland. For each participating household, the recycling rate was calculated. After that, the average household was determined. The obtained figures were summarized in Figure 1, which shows the average recycling rate, i.e., 69.0% ± 2.7%.

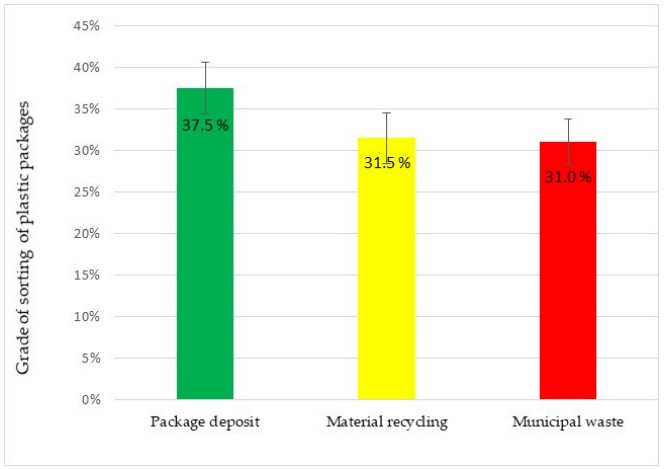

**Figure 1.** Plastic packages are divided into sorting fractions.

The average sorting by households is given with a 95% confidence interval (deposit and recycling). The reporting is based on the size and number of households as well as the age of respondents. The value of one-person households was 56%. The values for two-person and three-person households were 65% and 68%, respectively. For four-person households, the value was 63% and 57% for bigger households (Figure 2a). Considering the confidence interval, the material recycling activity in single-person households seems to be lower than in other households (Figure 2a). However, the *t*-test does not show statistical significance. The significance values are $p = 0.158$ (>0.050) for two-person households and $p = 0.578$ (<0.050) for three-person households.

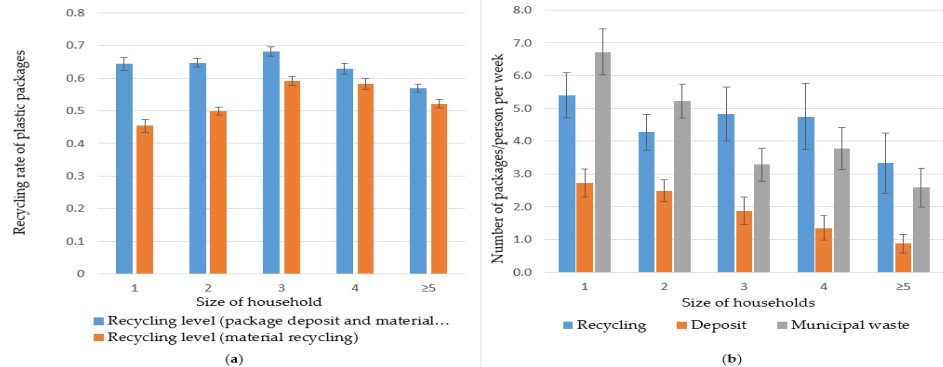

**Figure 2.** (**a**) The recycling rate of plastic packages with and without deposit in various-sized households; (**b**) The impact of household size on the recycling rate of food packages.

In this study, the impact of the number of packages per person per week in various-sized households was also investigated. Figure 2b shows the breakdown of packages/person in

three recycling groups. In households with three persons, the statistical significance was $p = 0.0002$ (<0.001), and with four persons, $p = 0.0017$ (<0.050). The number of discarded packages per person decreases because these households use large packages, e.g., big soft drinks bottles. These values were determined through the *t*-test.

The number of deposit returns per person also decreases as family size increases (Figure 2b). The difference between one- and three-person households is statistically significant, i.e., $p = 0.041$ (<0.050). Furthermore, the difference between two- and four-person households are statistically very significant, i.e., $p = 0.000005$ (<0.010). The large number of packages/person in two-person households may be because more food is prepared at home in these households than in single-person households. Two-person households clearly produce more packages/person than other households. They normally buy packages in sizes of either 330 mL or 500 mL.

## 3. Discussion

The purpose of food packaging is to protect the product from external hazards. Conventional packaging materials, e.g., metal, glass, and polymers, act as a barrier for microorganisms, but their sealant is a food safety risk. The main external causes of microbial contamination of heat-processed foods are contamination through the air, water, unfilled thermal polymers, and package damages. When the PET packages are extruded into food contact packages, these packages are normally not a safety issue, except for water bottles, which have been stored at room temperature. Any interaction between the packaging material and the food is undesirable because the interaction may have either toxicological or microbiological effects. Furthermore, the consumer may reduce the shelf-life or the sensory quality of the product by using the packages wrongly [3]. The food safety aspect must thus be considered when recycling material for use in food packages.

Poças et al. [27,28] collected data during a 30-day period from 34 households with 105 consumers. The amount of consumed food was 2982 kg, which included 303 kg of packaging material. The packaging is now lighter than before. The reason for this is that more plastics are used instead of glass. Another reason is that today, the weight of a plastic bottle is 50% lighter than in the 1970s [29]. The weight of the 0.5-L plastic bottle can be estimated to be approximately 0.025 kg. Approximately 0.3 kg plastics/person/week is formed, based on information in this study.

Sorting plastics can be confusing and uncomfortable, and therefore, consumers may choose to leave out sorting plastic packages. In this study, the number of packages was dimensioned by quantity to increase the recycling result. Alternatively, the packages could have been weighed. Tracking is a good way to train people in recycling material, but tracking is quite difficult for plastics. The quantification was chosen during this 1-week monitoring period to increase the success rate. In the participating households, the recycling rate for plastic packages was 61%, which exceeds the EU target of 55% for 2030 [1,2]. It enables Finnish households to fulfil the expected recovery rate [30].

Packaging materials can be used to reduce both food waste and environmental impact. Increasing the recycling of plastic waste is included in the European environmental policy to reduce both environmental impacts and dependency on resources outside the EU region [17]. The EFSA Panel on Food Contact Materials, Enzymes, Flavorings and Processing Aids [31] launched opinions on recycled PET in 2017. Nemat et al. [12] referred to the waste plan in Sweden for 2017–2020, in which the recycling rate in Sweden was 65%. The Swedish target is to improve this rate further, which can be enabled through easy and proper sorting and separation instructions on the plastic packages [12], i.e., the packages can either be reused after rinsing or the material recycled into new packages. It is to be mentioned that the Dutch plastic packaging waste recycling value chain has failed to meet the full net recycling rate of pure polymer packages of recycled material [11]. The purity of returned plastic food packaging is a challenge. An average polymer purity of 97% for recycled plastic products is needed to achieve a 72% net recycling rate. Plastic packaging consisting of multilayer laminates (O) is the most difficult of the packaging materials to exploit when a

single polymer is easily recyclable. The skills in identifying different plastic grades should belong to the basic education in all countries. In cases when the consumer is incapable of evaluating polymers worth recycling, depositing is an effective way to direct the recycling of material. This study showed that only 5% of the PET packages ended up in waste, which is most probably affected by the Finnish deposit system (Table 2). The recycling system is not a guarantee that materials reach the recycling point. Thus, enhancing packaging material recycling must also be highlighted in the future.

Reichert et al. [17] estimated that bio-based materials could replace some multilayer packages used specifically to adapt gas barrier properties and tune biodegradability. A key issue in the design of biodegradable packages is either to reduce food waste or to increase the processing of bio-waste [32]. With improved packaging functions, users' needs are met, and with new habits, it is likely that the amount of food waste will be reduced. According to Wikström et al. [10], research and expert workshops have shown that smaller packages and better information about the safe use and storage of food affect the reduction of food waste. Costello et al. [8] evaluated eleven waste management strategies used in the waste reduction model. The above studies show that sustainability in the recycling of packages must be improved on global, regional, and local levels. The industrial transition to a more sustainable use of plastic material has already begun, but the process will take years to become feasible both in industrial operations and in the economy.

## 4. Materials and Methods

It was decided to quantify the number of packages in order to obtain reliable results even though the recycling target in the directive (EU) 852/2018 [1] is based on weight. The task was compulsory for all students in one of the courses. In the other course, the task was to monitor either food or packaging waste. The students reported both the number and type of food packages during the one-week monitoring period in their households using a formal Excel sheet. The packaging material monitored in the households was grocery packages, and the items were classified into 17 types of packages. In this study, six types of plastic polymer packages were used. These packages were reported according to symbols in the international coding system. Images of the most common categories of packages and international recycling codes were added to the Excel sheet to facilitate the monitoring (Figure 3). Otherwise, there would have been an obvious risk that the students filling in the Excel sheets would identify the plastic material incorrectly. The students marked the end-use for each package separately. In case the cover was of a different material than the rest of the package, the response contained two separate entries. Furthermore, the number of packages was classified based on household size and age of the members.

| | Plastics | | | | | |
|---|---|---|---|---|---|---|
| Plastic Recycling Code Type of package | 1 PETE bottle/tray | 02 PE-HD bottle/tray | 04 PE-LD wrapping/bag | 05 PP tray/beaker | 06 PS tray/jar | 07 O tray&lid |
| Picture of Example Package | | | | | | |
| Target/Amount | pcs | pcs | pcs | pcs | pcs | pcs |
| Deposit | | | | | | |
| Recycling | | | | | | |
| Municipal waste | | | | | | |

**Figure 3.** The Excel sheet for reporting plastic food packaging waste. The various packages/materials were divided based on how the packages/materials were recycled, deposited, or placed in municipal waste.

The various packages/materials were divided based on how the packages/materials were recycled, deposited, or placed in municipal waste. The data were visualized using graphs based on an average and an error margin of 95% at the confidence level (shown in Figures 1 and 2). The background variable was the household size. The statistical significance of the average samples obtained from the background variables was assessed using a *t*-test. The *t*-test has been performed using the incoherent variance (heteroscedastic) of the two samples.

## 5. Conclusions

The focus of this study was on households with at least one university student, and the main limitation was that no sampling method was used. Furthermore, participation in the study course(s) may have increased recycling activity. The recycling of plastic packages is tricky for households, although an international recycling label has been added to the plastic packaging. PE-LD, PS, and O respond to approximately half of the mass ending up in material recycling, even though they have a lower renewal value than PE-HD, PP, and PET. The EU's recycling target for 2025 is set to 50%; this can be hampered by the low purity of plastic food packages. According to this study, the recycling rate of plastic packages in households was 61%. The size of the households did not affect the recycling rate significantly (56–68%).

The balance between material recycling and exploiting energy is dependent on how much energy is captured and what the recovered energy replaces. Another dimension is that the size of households affects the volume of packages/person. Large households use bigger pack sizes and fewer packages than small households. Thus, there is a challenge for the food industry to optimize the sizes of the food product packages for the various consumer categories. There is also a challenge in limited retail shelf space, which is more obvious, particularly in sparsely populated market areas, e.g., in Finland. A target for further research would be to study the impact of different types of food packages, package sizes, and processing methods to reduce food waste as well as the environmental footprint. The limited sizes of packages affect both food and package waste.

One purpose of the performed study was to educate students through their own experiences. The lack of studies on plastic recycling at the university level was also noted by Bennett and Alexandridis [33]. Training in plastic recycling for students equips them with the knowledge and skills to make informed decisions as consumers and implement plastic recycling systems at a professional level. Thus, a greater benefit of this study shows that small households applying such knowledge can reduce their packaging waste. The obtained figures follow common information for Finnish circumstances [19], and it shows that Finnish households will be able to fulfil the EU limits [1,2,6].

**Author Contributions:** Conceptualization, J.A. and M.N.; methodology, J.A. and M.N.; formal analysis, J.A.; investigation, J.A. and M.N.; data curation, J.A., M.N. and G.W.; writing—original draft preparation, J.A.; writing—editing, J.A., M.N. and G.W.; writing—critical review, G.W.; visualization, J.A. All authors have read and agreed to the published version of the manuscript.

**Funding:** This work was financially supported by Seinäjoki University of Applied Sciences (SeAMK) and the Finnish Ministry of Education and Culture.

**Data Availability Statement:** Not applicable.

**Acknowledgments:** Seinäjoki University of Applied Sciences approved the work to be performed using working hours.

**Conflicts of Interest:** The authors declare no conflict of interest.

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
