# Peer review of "Recycling of Plastic Food Packages: A Case Study with Finnish University Students"

_recycling, doi:10.3390/recycling8010023_

Round 1

Reviewer 1 Report

Recycling and reducing plastic packaging waste is still a topic without sound solution.

Here are my comment to the authors with a reference to the line number:

9&12 - the term 'wasted' is unclear, please specify. Discarded?

28 - 'Therefore, the recycling of..' should be rephrased, as the norm states " .. should take measures to ensure recycling of such packaging.'

56 - use/convert to SI units

59 - 'bio-based polymers belong to polyesters' is not factual, e.g. bio-PE (even though the reference might state stat)

78 - '..caused by packaging that recycling of scrap packaging materials could give reasonable environmental benefits.'  rephrase, it is unintelligible currently.

96 - 'The recycling plants have evolved significantly.' You could also mention different plastics recycling methods and their possibilities here as you do mention pyrolysis in L144.

Regarding the Results and the Discussion sections, add some estimate to the study regarding the amounts of the packaging waste in kg.

However, these findings identify how many packages an individual has to process, which might directly affect the willingness for sorting and recycling. Bu as such, it is very difficult to compare to other papers.

Author Response

Dear Reviewer

Thank you for giving us the opportunity to submit a revised draft of our manuscript titled Recycling of Plastic Food Packages: A Case Study with Finnish University Students to Recycling. We appreciate the time
and effort that you and the reviewers have dedicated to providing your valuable feedback on our manuscript.

We are grateful to the reviewers for their insightful comments on our paper. We have been able to incorporate changes to reflect most of the suggestions provided by the reviewers. We have highlighted in blue color the changes within the manuscript.

As overall comments, the changes leading to new text can be seen with highlighting in blue. Changes leading to deletion of text are visible as red text crossed out.

The authors

9&12 - the term 'wasted' is unclear, please specify. Discarded? Discarded is a clearer definition, which we have used.

28 - 'Therefore, the recycling of ...' should be rephrased, as the norm states " ... should take measures to ensure recycling of such packaging.'  This has been redrafted. L31-33

56 - use/convert to SI units. We have converted the mass into the SI unit kg. L60

59 - 'bio-based polymers belong to polyesters' is not factual, e.g., bio-PE (even though the reference might state stat) This has been changed. Furthermore, we have made other changes e.g., polyester-type bio-based polymers biodegrade slowly. L63-64

78 - '… caused by packaging that recycling of scrap packaging materials could give reasonable environmental benefits.'  rephrase, it is unintelligible currently. The reference has been deleted as a too common source. L82-84

96 - 'The recycling plants have evolved significantly.' You could also mention different plastics recycling methods and their possibilities here as you do mention pyrolysis in L144. Here, a reference with text has been added: L101-104

Regarding the Results and the Discussion sections, add some estimate to the study regarding the amounts of the packaging waste in kg. An estimate is presented on lines L247-251.

However, these findings identify how many packages an individual has to process, which might directly affect the willingness for sorting and recycling. It is very difficult to compare this to other scientific papers, because we were not able to find any comparable paper for this quantity.

Reviewer 2 Report

The paper is interesting and should be published.

The Summary, Introduction and Conclusion should be improved by:

-          « Consumers may choose not to sort packages, because sorting can be confusing and uncomfortable. » : the modern case where all packaging (cardboard, metals, plastics of all types) are collected in one bin and sorted in facilities is not presented and discussed, at least from literature data. Is it better/realistic to improve the collection rate that every consumer becomes a plastic specialist? Use the Dutch case (p 7) to feed the discussion

-          Please make a statistical comparison of the means that you discuss (distribution normal or not; t-test or ranked test…) 

Minor corrections:

Introduction

P 2 l 83 « de-trade » ?

M&M

P 8 l 273 The caption of Figure 4 is repeated in the text. Explain in the caption plastic “O”

Results

P 3 Table 1 Explain “target group by household size”

P 5 l 172 “The sorting The average by households is given with 95% confidence interval. (deposit and recycling) of nascent plastic packaging waste depends on the household size.” Please rephrase.

P 6 l 224 “which is due to the structure 224 of sub-crystalline plastics is destroyed during melting” Please correct

Author Response

Dear Reviewer

Thank you for giving us the opportunity to submit a revised draft of our manuscript titled Recycling of Plastic Food Packages: A Case Study with Finnish University Students to Recycling. We appreciate the time
and effort that you and the reviewers have dedicated to providing your valuable feedback on our manuscript.

We are grateful to the reviewers for their insightful comments on our paper. We have been able to incorporate changes to reflect most of the suggestions provided by the reviewers. We have highlighted in blue color the changes within the manuscript.

As overall comments, the changes leading to new text can be seen with highlighting in blue. Changes leading to deletion of text are visible as red text crossed out.

The authors

« Consumers may choose not to sort packages, because sorting can be confusing and uncomfortable. » : the modern case where all packaging (cardboard, metals, plastics of all types) are collected in one bin and sorted in facilities is not presented and discussed, at least from literature data. Is it better/realistic to improve the collection rate that every consumer becomes a plastic specialist? Use the Dutch case (p 7) to feed the discussion

In this report we are focusing plastic packages. We have planned to publish a review with all types of packages in a separate manuscript.

-          Please make a statistical comparison of the means that you discuss (distribution normal or not; t-test or ranked test…); The significance of the difference between the different household averages has been assessed on the basis of the p-value obtain through a t-test. The values have been added

Minor corrections:

Introduction

P 2 l 83 «de-trade» ? compensation L89

At revising the manuscript, we have used the word “Discard”.

P 8 l 273 The caption of Figure 4 is repeated in the text. Explain in the caption plastic “O”
The labelling O or 07 stands for mixtures of other plastics and other plastics. L192-194

Results

P 3 Table 1 Explain “target group by household size”
This tells how many people belong to a household of a certain size.

P 5 l 172 “The sorting The average by households is given with 95% confidence interval. (deposit and recycling) of nascent plastic packaging waste depends on the household size.” Please rephrase.
This determination of the mean error limit has been transferred to the methodology section.

 P 6 l 224 “which is due to the structure 224 of sub-crystalline plastics is destroyed during melting” Please correct.
This claim is based on the empirical experience of the users of recycled plastic and is therefore removed.

Reviewer 3 Report

This study investigated the effects of the size of the household on the volumes of packages/persons as well as the recycling rate of these plastics. The results obtained from this study are interesting however the writing, as well as the result presentation, are really terrible. The whole manuscript needs to be rewritten and rearranged. The obtained information needs to be discussed in more detail in the results&discussion as well as conclusions parts. To improve the manuscript, please revise it by following the comments and suggestions below.

- The abstract needs to be rewritten. It needs to include all information, objectives, scope, methodology, results, discussion, and conclusion.

- Please check the main story of the article. Why do results and discussion come before materials and methods? Please check other published articles and follow them.

- To make more attraction for the readers, the authors must identify the objectives and expected benefits more clearly. Why this study is needed?  What is the benefit of this study?

- The introduction does not let the readers understand this article's objective. The authors should revise the introduction following these instructions: background, literature review, research gap, purpose or objective, and the expected outcomes.

- The table must follow the template of the MDPI journals.

- Check the reference format.

- 2. Results > 3.1 ????

- Line 114-127, the results need to be discussed in more detail.

- Line 134-136, is the plastics listed in order?

- Line 142-145, the discussion using the information from published papers needs to be written smoothly, not only put it.

- The figures need to be revised. Please carefully checks the error bars, and axis (numbers and digits).

- Figures 3a and 3b: These figures should be together.

- Please check the captions of Figures 3a and 3b. The second sentence of each of them should be the notes, not the captions.

- Line 172-177, please carefully check. There are many typos.

- Discussion, the discussion using the information from published papers needs to be written smoothly, not only put it one-by-one.

- Line 254-256, this sentence is presented too many times. It is better to clarify it earlier in the introduction.

- Please check and revise the figures, the explanation could be put in the main text. Do not put too many details in the captions but please put them in the main text.

Author Response

Dear Reviewer

Thank you for giving us the opportunity to submit a revised draft of our manuscript titled Recycling of Plastic Food Packages: A Case Study with Finnish University Students to Recycling. We appreciate the time
and effort that you and the reviewers have dedicated to providing your valuable feedback on our manuscript.

We are grateful to the reviewers for their insightful comments on our paper. We have been able to incorporate changes to reflect most of the suggestions provided by the reviewers. We have highlighted in blue color the changes within the manuscript.

As overall comments, the changes leading to new text can be seen with highlighting in blue. Changes leading to deletion of text are visible as red text crossed out.

The authors

This study investigated the effects of the size of the household on the volumes of packages/persons as well as the recycling rate of these plastics. The results obtained from this study are interesting however the writing, as well as the result presentation, are really terrible. The whole manuscript needs to be rewritten and rearranged. The obtained information needs to be discussed in more detail in the results&discussion as well as conclusions parts. To improve the manuscript, please revise it by following the comments and suggestions below. The authors have rewritten the manuscript paying regard to all reviewers’ comments.

- The abstract needs to be rewritten. It needs to include all information, objectives, scope, methodology, results, discussion, and conclusion. The abstract has been redrafted based on comments of the reviewer.

- Please check the main story of the article. Why do results and discussion come before materials and methods? Please check other published articles and follow them.
Firstly, the authors have followed the template. Secondly, after the review, the corresponding author checked several Recycling-articles and based on this we changed the order of the chapters.

- To make more attraction for the readers, the authors must identify the objectives and expected benefits more clearly. Why this study is needed?  What is the benefit of this study?
According to the writers, a consumer acts as an operator at the first intersection of recycling. The consumer is interested in the recycling system. It plays an important role that the system works. The recycling system processes individual pieces, it does not work in kilograms of mass, but success is assessed on the basis of mass flows.

- The introduction does not let the readers understand this article's objective. The authors should revise the introduction following these instructions: background, literature review, research gap, purpose or objective, and the expected outcomes.
The authors state that efforts have been taken to improve the readability of the introduction section.

- The table must follow the template of the MDPI journals.
The tables have been modified in accordance with the template.

- Check the reference format.
The format of the references has been corrected.

- 2. Results > 3.1 ????
The authors have followed the template. Now in the revised manuscript the order has been made more logical.

- Line 114-127, the results need to be discussed in more detail.
This part of the results has been rephrased. L 162-177

- Line 134-136, is the plastics listed in order?
The polymers are listed according to the order of material recycling labels. L 191-194

- Line 142-145, the discussion using the information from published papers needs to be written smoothly, not only put it.
The authors have rephrased the Discussion part. The rephrasing has been made visible through highlighting. L 197-200

- The figures need to be revised. Please carefully checks the error bars, and axis (numbers and digits).
The tables giving the figures have been checked.

- Figures 3a and 3b: These figures should be together.
The two figures have been compiled according to the comments obtained. An excellent call has been made.

- Please check the captions of Figures 3a and 3b. The second sentence of each of them should be the notes, not the captions.
The authors have acted accordingly.

- Line 172-177, please carefully check. There are many typos.
These mistakes have been corrected. L229-232

- Discussion, the discussion using the information from published papers needs to be written smoothly, not only put it one-by-one.
Big parts of the discussion have been rephrased.

- Line 254-256, this sentence is presented too many times. It is better to clarify it earlier in the introduction.
This text is now found in the introduction.

- Please check and revise the figures, the explanation could be put in the main text. Do not put too many details in the captions but please put them in the main text.
The authors have taken efforts in correcting the presented pictures.

Round 2

Reviewer 1 Report

L154:  . The data were analysed using graphs. ? What does this mean?

L194:  'Plastic Identification Codes' - please add reference to the codes.

L230: 'nacent' is rather strange term to use within the context.

Thank you for the hard work, but I still hope to see some minor improvements. Merry Christmas and better new Year!

Author Response

Thank you for giving us the opportunity to submit a revised draft of our manuscript titled Recycling of Plastic Food Packages: A Case Study with Finnish University Students to Recycling. We appreciate the time
and effort that you and the reviewers have dedicated to providing your valuable feedback on our manuscript.

We are grateful to the reviewers for their insightful comments on our paper. We have been able to incorporate changes to reflect most of the suggestions provided by the reviewers. As overall comments, the changes leading to new text can be seen with highlighting in yellow. Changes leading to deletion of text are visible as red text crossed out.

The authors

Reviewer 3 Report

The revised manuscript does not improve enough. Many parts are not clearly revised as per the reviewer's comment and suggestion. I decide to reject this manuscript since the authors could not revised the manuscript properly.

However, to improve your manuscript please find my comments and suggestion below:

- The abtract still not well presented. All information incluing the objectives, scope, methodology, results, discussion, and conclusion still not clearly explained.

- The main story of this manuscript still really complicate and lack of logic as well as story flow. Please try to read more published papers to improve your academic writing.

- Table 1 is not well organized.

- Finally, please reconsider my previous comments during the first reviewing process.

Author Response

(The authors gave the same response as above.)

Round 3

Reviewer 3 Report

Sorry to say that the manuscript still not good enough to be published. The authors should review more paper relating to the waste management. In my opinion, this manuscript still need to be improved a lot espectially a storytelling to make the readers understand it easily. Based on the revised manuscript and response to reviewer, please find my comments below.

The abstract still not well-presented. The first sentense is not commonly used in the abstract and it is also similar to the sentense in Line 14-16. The authors already put everthing that should appear in the abtract but it look like each sentence was put separately with out connection of each others. This problem was also found it the maintext espeatailly for introduction that cause this manuscript difficult to understand. The author should focus on the writing with good storytelling to make the flow of reading. Please carefully thing about readers, do not only to revise the manuscript only for answer the reviewers by following each comments and suggesstion separatelly. Whole manuscript need to be connected with good story and flow.

For the benefit of this manuscript, if the benefit is for educating the student though their own experience, it can be in the room class this benefit is not appropriate enough to be the benefit of the paper. The authors should focus on the greater benefit.

The materails and methods are find. But the results, discussion, and conclusions still not good, espectially conclusions. The conclusions should conclude and/or summarize the finding from this work not other works.

Author Response

Dear Reviewer

In this revision the authors of the manuscript have marked the changes using the “Track Changes” function. The answers from the authors are marked in red and blue in this version of the manuscript. The comments of the reviewer is given in black.

The Authors
